# An Intelligent Multi-Floor Navigational System Based on Speech, Facial Recognition and Voice Broadcasting Using Internet of Things

**DOI:** 10.3390/s23010275

**Published:** 2022-12-27

**Authors:** Mahib Ullah, Xingmei Li, Muhammad Abul Hassan, Farhat Ullah, Yar Muhammad, Fabrizio Granelli, Lucia Vilcekova, Tariq Sadad

**Affiliations:** 1School of Mechanical Engineering and Electronic Information, China University of Geosciences, Wuhan 430074, China; 2Department of Information Engineering and Computer Science, University of Trento, 38122 Trento, Italy; 3School of Automation, China University of Geosciences, Wuhan 430074, China; 4School of Computer Science and Engineering, Beihang University, Beijing 100191, China; 5Information Systems Department, Faculty of Management Comenius University in Bratislava, Odboj’arov 10, 82005 Bratislava, Slovakia; 6Department of Computer Science, University of Engineering and Technology, Mardan 23200, Pakistan

**Keywords:** IoT, smart services, monitoring, autonomic computing, facial recognition, voice recognition, robotics

## Abstract

Modern technologies such as the Internet of Things (IoT) and physical systems used as navigation systems play an important role in locating a specific location in an unfamiliar environment. Due to recent technological developments, users can now incorporate these systems into mobile devices, which has a positive impact on the acceptance of navigational systems and the number of users who use them. The system that is used to find a specific location within a building is known as an indoor navigation system. In this study, we present a novel approach to adaptable and changeable multistory navigation systems that can be implemented in different environments such as libraries, grocery stores, shopping malls, and official buildings using facial and speech recognition with the help of voice broadcasting. We chose a library building for the experiment to help registered users find a specific book on different building floors. In the proposed system, to help the users, robots are placed on each floor of the building, communicating with each other, and with the person who needs navigational help. The proposed system uses an Android platform that consists of two separate applications: one for administration to add or remove settings and data, which in turn builds an environment map, while the second application is deployed on robots that interact with the users. The developed system was tested using two methods, namely system evaluation, and user evaluation. The evaluation of the system is based on the results of voice and face recognition by the user, and the model’s performance relies on accuracy values obtained by testing out various values for the neural network parameters. The evaluation method adopted by the proposed system achieved an accuracy of 97.92% and 97.88% for both of the tasks. The user evaluation method using the developed Android applications was tested on multi-story libraries, and the results were obtained by gathering responses from users who interacted with the applications for navigation, such as to find a specific book. Almost all the users find it useful to have robots placed on each floor of the building for giving specific directions with automatic recognition and recall of what a person is searching for. The evaluation results show that the proposed system can be implemented in different environments, which shows its effectiveness.

## 1. Introduction

One of the key technologies utilized in applications for location-based services such as augmented reality (AR), the Internet of Things (IoT), artificial intelligence (AI), robotics navigation, and consumer analytics, is positioning [1,2]. Currently, outdoor GNSS-based smartphone locating services are capable of centimeter-level precision. Using the current indoor positioning systems, it is still challenging to attain low-cost, reliable, and strong indoor positioning effects because GNSS signals are not available in an indoor environment [3]. A key indoor positioning technique in this field is the vision-based indoor positioning of smartphones, which uses real-time ornamental texture data in a space and does not require extra consumption of resources to transform the indoor environment. Visual localization has recently attracted a lot of attention in the realm of indoor navigation [4]. To address the issue of image-based localization, the majority of cutting-edge techniques rely on regional features like SIFT or SURF [5]. These techniques typically involve two steps: perspective-n-point, which determines the extrinsic characteristics, and descriptor match, which establishes 2D–3D matches between characteristics derived from the positioning image and 3D points.

With the advancement and development of the architecture of intelligent systems, indoor navigation systems are becoming more important [6]. In any modern society, to help people reach their desired destination or achieve their required goals hassle-free and in a timely manner, there is a need for a navigational system. Currently, to help people navigate through large buildings or complex infrastructure, multiple systems have been developed based on ultrasonic sensor positioning, WiFi, RFID localization, etc. [7]. Among the existing systems, most of them target academic buildings, while some of them are designed for specific buildings such as shopping malls, grocery shops, etc., using indoor mapping techniques. The main drawback of these systems is that they are not generalized and cannot be used for varying situations occurring in different buildings [8]. To cope with the current indoor navigational problems, researchers are now focusing on image processing-based intelligent solutions that can automatically track a person and guide him/her through specific paths to their targeted areas [9,10]. There are numerous research works on indoor navigation, but the majority of them focus on a single area or feature and incorporate all of the necessary features for ease of access.

In this paper, we present a novel idea for an intelligent multi-floor navigational system that can be implemented in a variety of scenarios depending on the environment and can be changed based on location. In this study, we mainly focused on the deployment of multiple robots on each floor of a building that communicate with each other and with the person who needs navigational assistance. Once a person requests navigation from a single robot, each one gets updated and gives automatic directions after recognizing the person using facial recognition algorithms. The main features we used in the proposed system are: first, speech to get the required input from a person who wants to navigate in the environment; and then converting it into text in order to make it readable for the algorithm. The second feature is facial recognition, which ensures the automatic recognition of a person without giving any further commands once registered. The third feature is using voice broadcasting to communicate with the person according to their navigational requirements. All of the mentioned features are merged into a single mobile Android application that can be attached in the future to a mobile or stationary robot for more interactive communication and navigation. Before finalizing the required application, we started by taking a survey of users for the requirement of multi-story building navigation. The first step is to find out the percentage of people who face problems or need assistance in a multistory building, as shown in Figure 1. From Figure 1, it is clear that more than 85% of people faced problems; 37 people were selected for the random response of the questionnaire survey.

Furthermore, according to the responses of 37 people from different groups, in multistory buildings, they mostly face difficulty finding a specific location, and sometimes they need assistance for entry and exit points, as shown in Figure 2 and Figure 3.

The main and most important drawback of the earlier indoor navigation systems is that they were developed only for a particular location or building. When it comes to different locations and buildings, these systems are not applicable. In order to overcome this issue, we developed an indoor navigational system that is generalized and can be implemented in different environments or can be changed according to the environment. The main contributions of this study are given as follows.

We proposed a model of multi-story library navigation based on multiple robotic models using Android phones and tablets as communication platforms, where once a user enters his or her required book area or library, he or she is guided automatically on each floor of the library;The proposed system consists of three basic and important modules (image recognition, speech recognition, and voice broadcasting). The developed system uses image processing techniques to recognize a person, speech recognition to get the user’s requirements, and voice broadcasting to help navigate on each floor of the building. In the proposed system model, a user gets registered using an Android app, where basic personal data are collected from him or her with real-time images. The real-time images are then used for face recognition to authenticate him or her on each floor of the building. After getting registered, the user is directed to the robots for the required book, and until the task is completed, the user is given directions on each floor. The proposed system model is generalized and can be implemented in any indoor environment; andThe results were attained by conducting a survey of the application users at the university library. For this purpose, we added 37 students’ data to the application and then recognized them separately to find out the level of accuracy of the image recognition algorithm. For speech recognition, Google voice-to-text is used and voice broadcasting is used to give proper suggestions/guidelines by entering multiple inputs and getting the suggestions from robots.

The remaining paper is organized in the following order; Section 2 represents the review of literature in the domain of indoor navigation systems. Section 3 illustrates the methodology adopted in the accomplishment of this research study. The experimental and simulation results are discussed in more detail in Section 4 of the paper. Finally, Section 5 concludes the overall theme and findings of this research study.

## 2. Literature Review

With the speedy advancement of photogrammetric, object recognition, and optical imaging technology, it is now possible to acquire images quickly and affordably, extract and correlate precise and effective picture features, and quickly solve the projection matrix and other exterior orientation-related problems [11]. Picture-based visual positioning provides the advantages of improved precision, context-rich knowledge, and excellent visual impact. Further, it has the potential to offer an affordable and precise active indoor positioning solution. As a result, worldwide academics have extensively examined visual positioning technologies. Besides the advantages and extensive use of different indoor positioning techniques, these techniques have some limitations as well, which are discussed by different researchers in their research studies. For example, Lluvia et al. [12] stated that indoor navigation faces various problems such as mapping the environment, indoor positioning, and planning for trajectory. Further, ISO 17438-1:2016 stated that indoor positioning and mapping are subcategories of navigational applications [13]. In general terms, navigation can also be referred to as a multi-field application within others that include asset management, mapping, tracking, localization, and so on. In our literature review, we came to the conclusion that indoor navigation, mapping, and indoor positioning are the most fundamental technologies, and their requirements change according to the circumstances. We divided the literature for this study into multiple subcategories that include indoor navigational systems, indoor mapping, face detection and tracking, speech recognition, and voice broadcasting.

Nowadays, most of the research work in the field of indoor navigation is based on image processing technologies. Numerous methods and systems have emerged as a result of the use of these technologies in numerous fields. For example, in reference [14], an indoor navigation system for blind people is developed using a mobile application in which color patterns around the application user are detected using image processing techniques, thereby providing assistance in the indoor navigation system. Indoor navigation is classified by researchers into several categories, such as computer vision, which includes omnidirectional cameras, 3D cameras, or inbuilt smartphone cameras for face and environment detection [15]. Wi-Fi, Bluetooth, and RFID are widely used communication technologies, and the most widely used method for navigation is the pedestrian dead reckoning technique, which uses the inbuilt sensors of smartphones and does not require external hardware [16]. Indoor navigation for blind people has been taken to the next level by B. Li et al. [17], who developed a solution for dynamic environment navigation using image processing techniques. Their proposed prototype contained a Google Tango mobile device, a smart cane with a keypad, and two vibration motors. Other than that, SLAM-based solutions for indoor navigation are presented in [18,19]. Similarly, in another study, for indoor navigation, communication technology such as WI-FI is used to navigate and localize mobile communication with different nodes, and based on the time taken for a response, location is detected [20]. Machine learning-based algorithms are also used for WIFI fingerprint-based tracking, and the results produced by these algorithms are more promising using SVM [21], neural networks, and KNN [22], where they specifically presented a technique for navigation in mines using fingerprint matching.

Most of the work also uses computer vision technology, which can be divided into two sub-research fields that use either an infrastructure of static cameras to track mobile entities (e.g., people, robots) or cameras attached to the mobile entities [23]. After advancements in camera technology and the development of advanced algorithms, the use of a mobile phone camera (with image processing techniques embedded in the camera, such as different filters), is being considered. For example, M. Li et al. [24] proposed multiple mobile application-based solutions using a smartphone, such as a precision single-image-based indoor visual positioning method in which the mobile camera color patterns were matched to find out the current position. Similarly, another approach was proposed in [5] for spatial visual self-localization using mobile platforms in urban settings, which showed promising performance in investigating the high-precision visual placement of cellphones in outdoor settings. References [25,26,27,28,29] investigated the most recent methods for image position tracking, using DL and visual location-based techniques for SOTA image features, as well as cutting-edge technology for extracting image features and retrieving the required images based on the extracted features.

Furthermore, related research for mobile application-based library management or multi-story building has been accomplished on multiple levels, such as [25], which developed a smart voice assistant for the library based on the internet of things using a raspberry pi and a speaker module to assist people. The main problem associated with the earlier approaches is the single point of concern, i.e., they are not generic and are used for a specific point of interest. They are primarily designed for one environment and cannot be used in another. This is indeed a serious issue that needs to be addressed by developing a system that can be used in multiple environments under different circumstances. In order to sort out this problem, we proposed a system that can be used in different environments. We combined multiple existing solutions and proposed a single algorithm that can search, give directions using voice broadcasting, recognize people using the Android tensor flow library, convert voice-to-text using Google Voice to Text, and then obtain the required information. The proposed system is tested in the university’s multi-story library with the new idea of multiple robots communicating with each other and with a person who is searching to find a particular book in the library. The proposed system is anticipated to be of great assistance to persons who are new and do not know where a book of a particular discipline is located in the library.

## 3. Proposed Methodology

This section represents the approaches adopted and the materials used in the completion of this research study. For developing an efficient mobile or computer-based application, third-party libraries play a vital role in time minimization, providing an efficient interface, data management, algorithm development, and easy communication between multiple modules [26]. The developed intelligent multi-floor navigation system is based on multiple recognition systems, i.e., speech, face, and voice broadcast using an Android application. To develop an application for multistory buildings, we focused on a single solution by implementing our changeable application. The proposed system comprises two main Android application systems, i.e., the indoor robot system, which is an administrative app, and the indoor robot navigation system, which enables the interaction of robots with the users. The methodology adopted in this study is represented in Figure 4.

The two main Android applications developed in this study are discussed in more detail in the following subsections.

### 3.1. Indoor Robot System Administration

The main function of the administrative application is to gather data from different sources. The first step is to sign in to the administration application, then multiple options appear for the admin to configure the whole system, which can be changed according to the requirements. The administrative android application (indoor robot system app) consists of several sub-modules such as robot management, floor management, shelf management, book management, and member management, as shown in Figure 5.

Robot management is used to add or remove a robot from the system. For every floor, there is a robot, and the number of robots is equal to the number of floors. Every robot in the robot management module has a unique name and ID, which are used to guide the customer on each floor of the building. Figure 6a shows the backend of the robot management module.

The floor management module is used for the management of floors, i.e., how many floors are there in the building and to place robots on each floor. In this module we add floors to the system and then assign each a robot to a floor. The two main attributes of this module are floor name and robot selection. Figure 6b illustrates the floor management module of the indoor robot system application.

The shelf management module of the indoor robot system is used for the arrangement of books on each floor of the building. The number of shelves on each floor of the building can be increased according to the needs and requirements of the library. The main attributes in this module include shelf number and floor selection. The backend of this module is represented in Figure 7a.

The book management module of the indoor robot system is used to arrange books on each floor of the building. The main attributes of this module are: book name, author name, floor selection, and shelf number. The backend of this module is shown in Figure 7b.

Similarly, the last module of this application is the member management module, which is used to register the members that will have access to the books in the library. The attributes included in this module are: member first name, last name, email address, phone number, and date of birth. The backend of this module is illustrated in Figure 7c.

### 3.2. Indoor Robot Navigation System

The second android application of the proposed system, i.e., the indoor robot navigation application, is deployed on the robots and consists of a monitoring screen that enables the interaction of robots and users, as shown in Figure 8. The main purpose of this application is to provide a connectionless and easy access to information needed by a person. When the robot application is opened, there are two options for the user. The first one is the monitoring screen, which is used for navigational help, and the second one is logout, as shown in Figure 8.

In the monitoring screen, the front or back camera (depending on place of usage and administration requirements) of a mobile phone stays on until a face is detected and recognized using the Tensorflow Lite library for Android [27]. The first step of the monitoring screen module is to recognize the user using a face recognition algorithm. The robot will announce the name of the user if he/she is already registered in the system, as shown in Figure 9a. The monitoring system also recognizes the location of the user, such as on which floor he or she is. Once a person is identified, the next step is to obtain input from that person regarding the required book, in the case of a library. Input from the user can be in the form of voice (Figure 9b), which is converted to text using Google API services for voice to text, or if the person is non-vocal, he/she can just enter the required book name using the keypad of the screen (Figure 9c). The robot will assist the user in finding out the location of the book, i.e., on which floor the book is and on which shelf it is placed, as shown in Figure 9d.

After application layout and design, the next step is the integration of algorithms for face detection and recognition, speech recognition and broadcasting, and database development and maintenance (adding new users, updating the information of existing users, and deleting users from the system database).

#### 3.2.1. Image Recognition Module of Screen Monitoring

This module is used to recognize whether the user is an authorized person or not. After recognizing his or her face, this module further allows him or her to access different books in the library. Figure 10 illustrates the workflow of the image recognition module.

The image of the user is acquired with the help of an RGB camera and then passed to the Viola Jones algorithm, which detects different points on the face. After this, different features are extracted from the acquired image and passed to the classification algorithm, i.e., CNN. CNN classifies the image and determines whose image it is; the user’s registration status is then checked. If the user is a registered user, then he or she is authorized to have access to each floor and book of the library. If the person is not a registered user, his request to use the library is denied.

#### 3.2.2. Voice Recognition Module of Screen Monitoring

After the user recognition, the voice recognition module allows the user to search for a particular book. If the book is in the library, the user is provided complete information about the book, i.e., book name, author name, floor number on which the book is place, and shelf number in which the book is placed. Figure 11 shows the workflow of the voice recognition system.

In this module, the user is asked to record their voice and search for a particular book. As the voice is in real-time, there is some noise. The noise is removed from the input voice by applying noise-removing filters. After removing the noise from the input voice, it is passed to the MFCC algorithm to extract useful features from it. After extracting useful features from the input voice, they are passed to the classification algorithm, i.e., CNN, which classifies the input voices and decides whether the book name for which the user has input the voice is in the library or not. If the book name is present in the library book list, then the person is informed about the complete information of the book, i.e., on which floor it is and on which shelf it is placed. If the book’s name is not in the library book list, then the person is informed that the book is not in the library list.

In this study, different deep learning and machine learning libraries are used for face detection and voice recognition. For example, for face detection and recognition, the TensorFlow Lite library for Android is used, which is a lightweight version of the TensorFlow library and is playing a vital role in embedded and mobile systems. The main feature of TensorFlow Lite is that it enables machine learning interfaces with small binary sizes and low latency. TensorFlow Lite also supports hardware acceleration with the Android neural network API. TensorFlow Lite applies many techniques for achieving low latency, including optimizing the kernels for mobile apps, pre-fusing activations, and quantized kernels that allow smaller and faster (fixed-point math) models. Another API that we used in our application is voice-to-text conversion [28] for user input of required items in our case book and text-to-voice conversion for specific directions of queries entered by the user. To search for the required book, we used another function, i.e., the event change listener [29].

## 4. Result and Discussion

This section represents all the experimental work accomplished in the conduction of this research work. To track the performance of our proposed indoor navigation system, we used two types of evaluation systems. The evaluation systems we used included system evaluation and user evaluation in order to enhance the functioning and performance of the proposed multistory navigation system.

### 4.1. System Evaluation

The first and most important step of a system before its implementation is its evaluation, to determine whether it meets the requirements of the topic for which it was developed or not. For the implementation and evaluation of the proposed system, we used an efficient architecture of a deep convolutional neural network to evaluate the performance of the system. The evaluation results reveal that using the proposed architecture, the performance of the system was superior both in terms of voice and face recognition. For training and evaluating the model, we accumulated 300 images of various individuals, of which 70% were used for training and 30% for testing. Parameter settings play a significant part in the performance of various ML and DL algorithms; therefore, we carried out several experiments in order to determine the optimal parameter settings for the proposed architecture of the convolutional neural network. Further, we also determined the optimal number of dense units for hidden layers based on the accuracy result. We also tested the dropout rate on a regular basis to evaluate the fit rate of our CNN by evaluating the accuracy of the results. As shown in Table 1, the highest accuracy value was obtained for facial recognition when we used a dropout rate of 0.3 and a dense unit of 113. Furthermore, the dropout rate we tested ranged from 0.1 to 0.5, increasing by 0.1, while the dense units were between 110 and 120. Figure 12 depicts the accuracy value as a function of the density unit number and the dropout rate.

Based on the accuracy results, we also tried to figure out the ideal number of dense units for hidden layers for voice recognition, which is pretty close to facial recognition. In order to assess the accuracy of the results and the fit rate of our proposed CNN, we also routinely tested the dropout rate. As shown in Table 2, the highest accuracy value was obtained for voice recognition when we used a dropout rate of 0.4 and a dense unit of 118. Indeed, the dropout rate we tested ranged from 0.1 to 0.5, increasing by 0.1 when the dense units were between 110 and 120. Figure 13 depicts the accuracy value as a function of the density unit number and the dropout rate.

Actually, Table 1 and Table 2 demonstrate the results of numerous experiments we conducted to determine the optimal number of dense units to use. In this study, we relied on the accuracy value to try out various values for the neural network parameters. We trained the classifier for voice and facial recognition during binary classification, and the model’s accuracy for both tasks was 97.92% and 97.88%, respectively. It is necessary to note that the prediction was made with the highest accuracy using the trained version of our proposed framework.

### 4.2. User Evaluation

The second step in evaluating the proposed model’s performance is user evaluation, which determines whether or not this system is beneficial to society. To do this, we conducted user evaluations of our proposed system. Keeping in mind the importance and need for a multistory navigational system, we conducted a comprehensive survey of application users in the form of different questionnaires. According to the survey report, more than 40% of the users get mobile application-based assistance, as shown in Figure 14. This proves the need for a solution consisting of mobile applications because people are already used to them.

The survey analysis results also prove that in the future, people will want indoor mapping and robot receptionists for multistory buildings, as shown in Figure 15.

Application graphical user interface was designed by keeping in mind ease of access and less interaction on screen by adding automatic face detection and speech recognition. Out of 35 responses, only 14% of users were not fully satisfied, while the remaining 86% of users gave a positive review of the GUI, as shown in Figure 16.

User registration in the administration application is also very simple, requiring only a facial picture, name, contact number, and email. Users’ responses were satisfactory, with most of them saying it was easy, as shown in Figure 17.

One of the key features of our application was the facial recognition of a registered user through the TensorFlow Lite library in the application. The application was tested in a university library, where the responses of users were also recorded. Face recognition accuracy was tested based on the responses of users who entered the library and went to the robotics application for help. Most of the users’ faces were detected efficiently. Face detection accuracy depends on lighting in the surrounding area, the face, camera angle, and the number of users registered.

For voice recognition, we used the Google speech recognition algorithm, which shows good accuracy in a quiet environment, and more than 70% of responses out of 35 entries were satisfactory. The sudden drop in score was due to surrounding noise and robotic device application microphone issues.

More than 50% of people were satisfied with the navigational help provided by a mobile application and successfully reached their required destination in a multistory building after taking input from robots placed on each floor. Responses recorded are shown in Figure 18.

Most of the people who tested this application responded that the solution provided is helpful in an indoor multistory building environment (Figure 19). Users also liked the idea of placing multiple robots on each floor because they could get directions whenever and wherever they needed them in an indoor environment (Figure 20).

The success of indoor navigation depended on achieving the targeted navigation in the library, and out of the 34 responses shown in Figure 21, more than 80% successfully navigated to a required book in the library while interacting with the provided solution. Lastly, based on the experience of users of our solution robotic application, 85% of the people recommended that the same idea can be implemented in other multistory buildings, as shown in Figure 22 and Figure 23, which proves that it is one of the best solutions for indoor navigational environments.

## 5. Conclusions and Future Works

With the continuous development of advanced technologies such as IoT, ML, DL, etc., an intelligent multi-floor navigational system using advanced identification techniques such as speech recognition, face recognition, and voice broadcasting based on Android applications is a new and interesting topic that needs to be investigated. In this study, we proposed an indoor navigation system that guides users to find a particular book on different floors of the library. The proposed system mainly consists of two Android apps, i.e., the administrative app (robot indoor system administration) and the navigation app (robot indoor navigation system). During the experimental process, facial and voice recognition were mostly accurate, but sometimes errors occurred due to environmental factors that can be further reduced. The proposed system was successfully tested for navigation in a multistory library. Further, with a few changes, the same work can be implemented in other indoor multistory buildings such as grocery stores, shopping malls, etc. The future work of this study is to implement the proposed system on more than five floors and to deploy more robots on each floor of the building. The proposed solution needs to be tested in other multi-story buildings for navigational help. Mobile phones can be replaced by moveable robots such as Pepper robots, which will make navigation more interactive in the future.

## Figures and Tables

**Figure 1 sensors-23-00275-f001:**
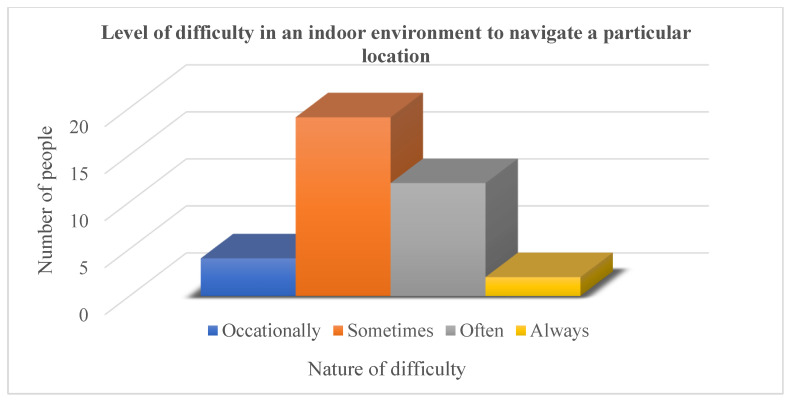
Level of difficulty faced by people for indoor navigation in multistory building.

**Figure 2 sensors-23-00275-f002:**
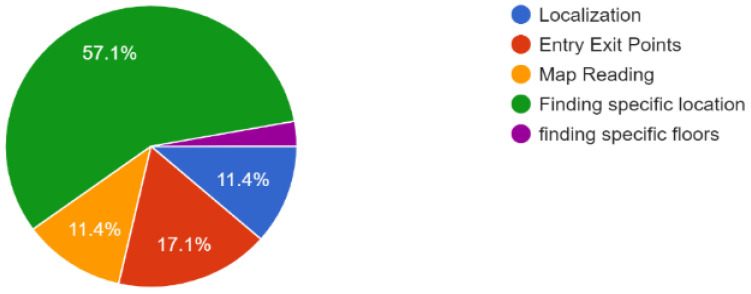
Issues faced by people in indoor navigation.

**Figure 3 sensors-23-00275-f003:**
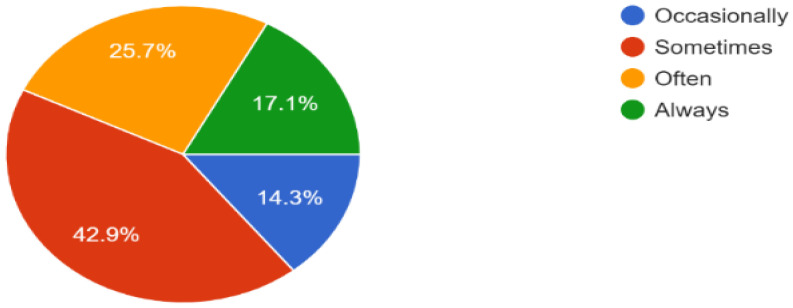
Percentage of help required by people for indoor navigation.

**Figure 4 sensors-23-00275-f004:**
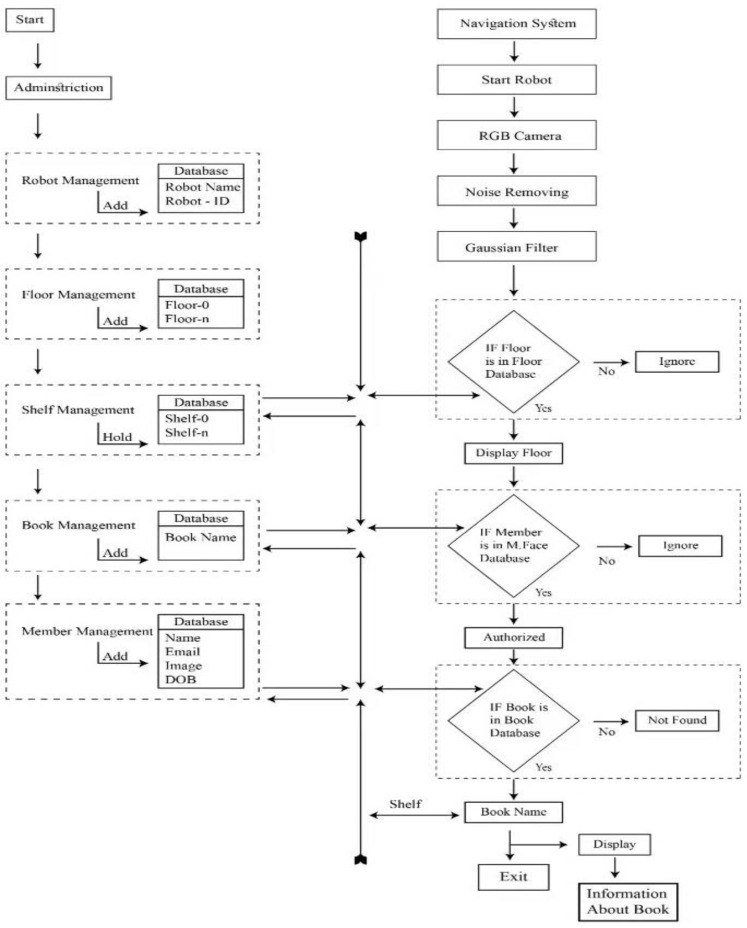
Methodology of the proposed system.

**Figure 5 sensors-23-00275-f005:**
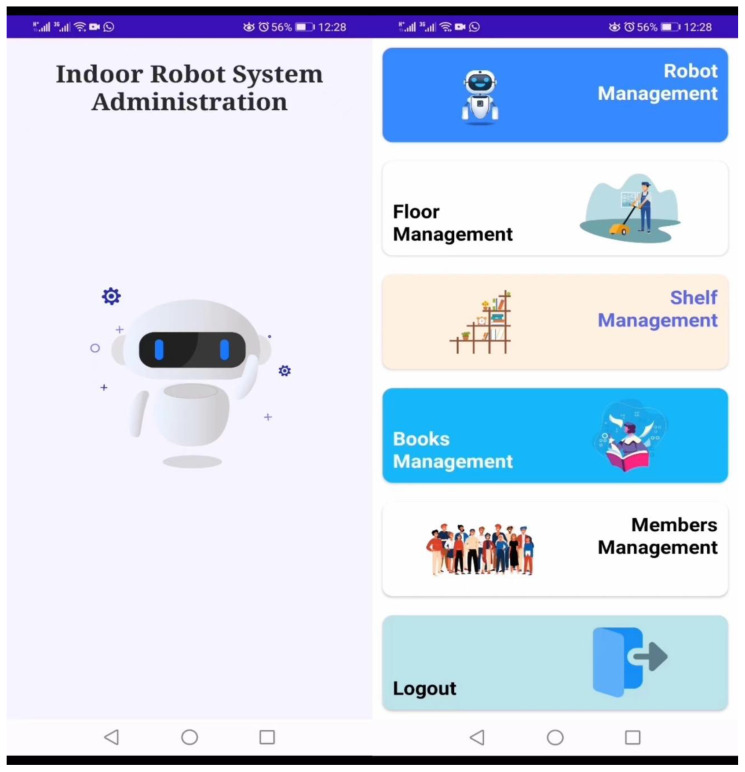
Indoor Robot System Application.

**Figure 6 sensors-23-00275-f006:**
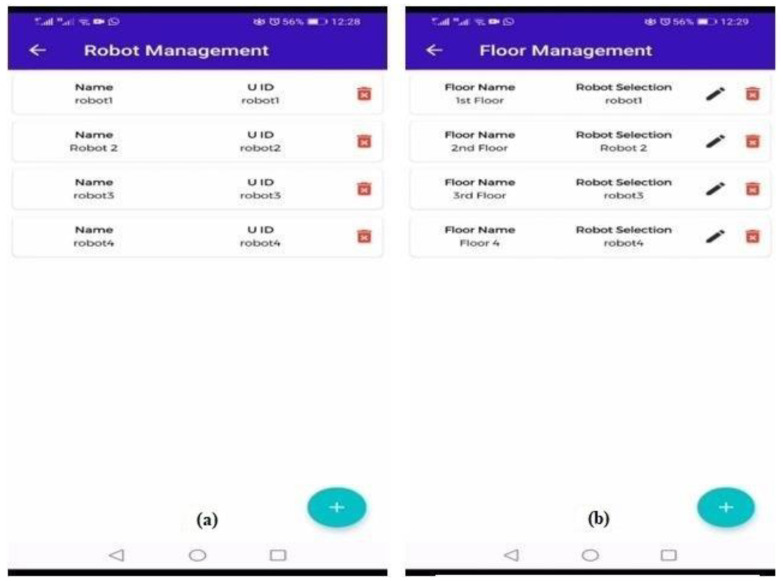
Sub-modules (Robot and Floor Management) of the Indoor Robot System App. (**a**) Robot Management. (**b**) Floor management.

**Figure 7 sensors-23-00275-f007:**
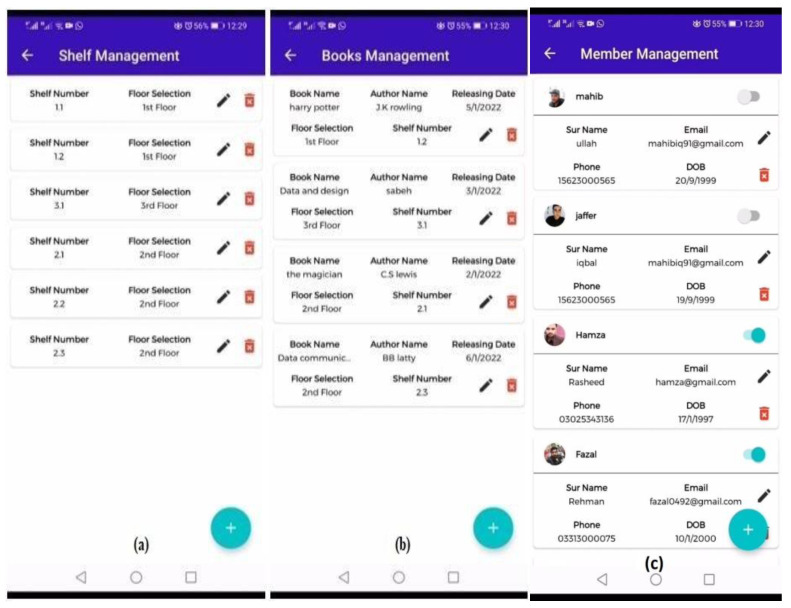
Sub modules (Shelf, Books, and Member management) of Indoor robot system app. (**a**) Shelf management. (**b**) Books management. (**c**) Member management.

**Figure 8 sensors-23-00275-f008:**
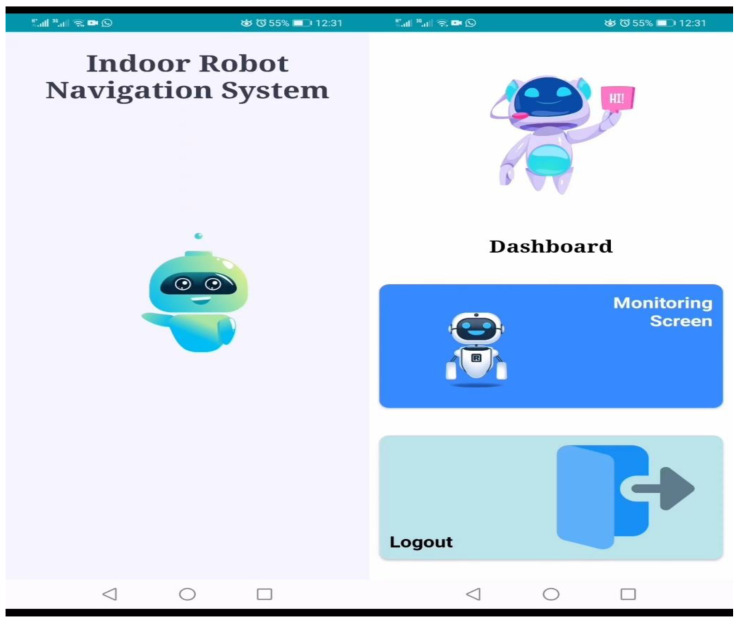
Indoor Robot Navigation Application.

**Figure 9 sensors-23-00275-f009:**
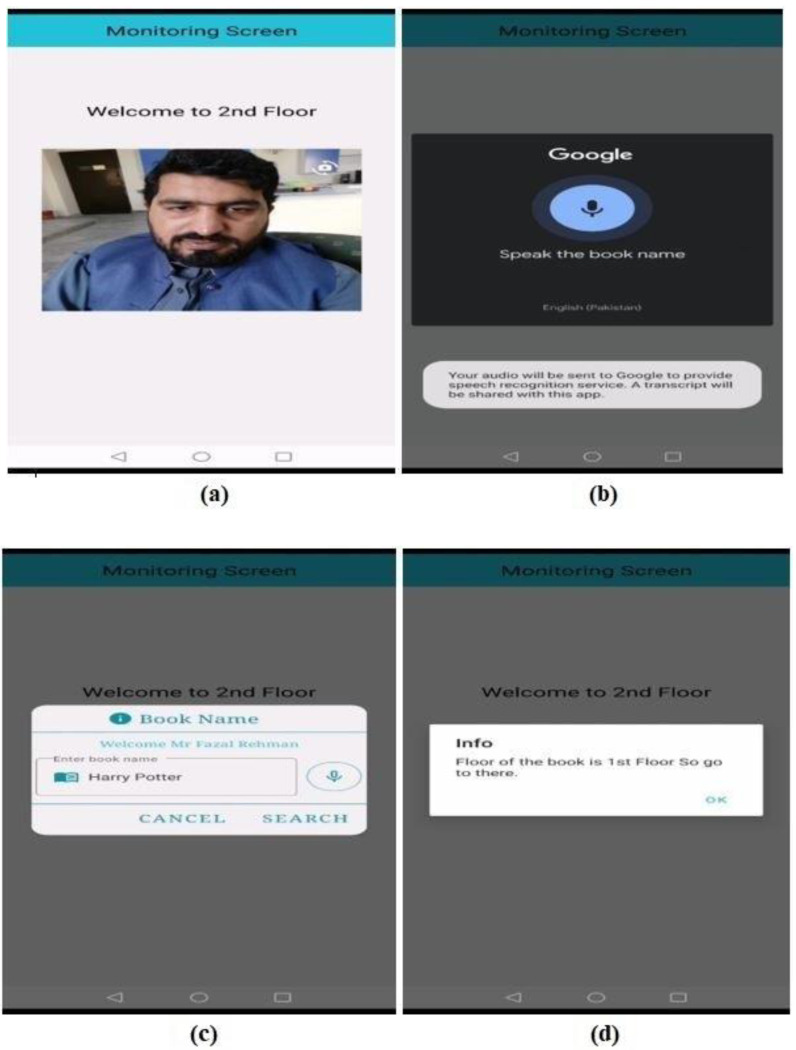
Indoor Robot Navigation Application Screen Monitoring. (**a**) Face/user recognition. (**b**) Speaking the book name. (**c**) Searching for the book name. (**d**) Suggesting floor of the book.

**Figure 10 sensors-23-00275-f010:**
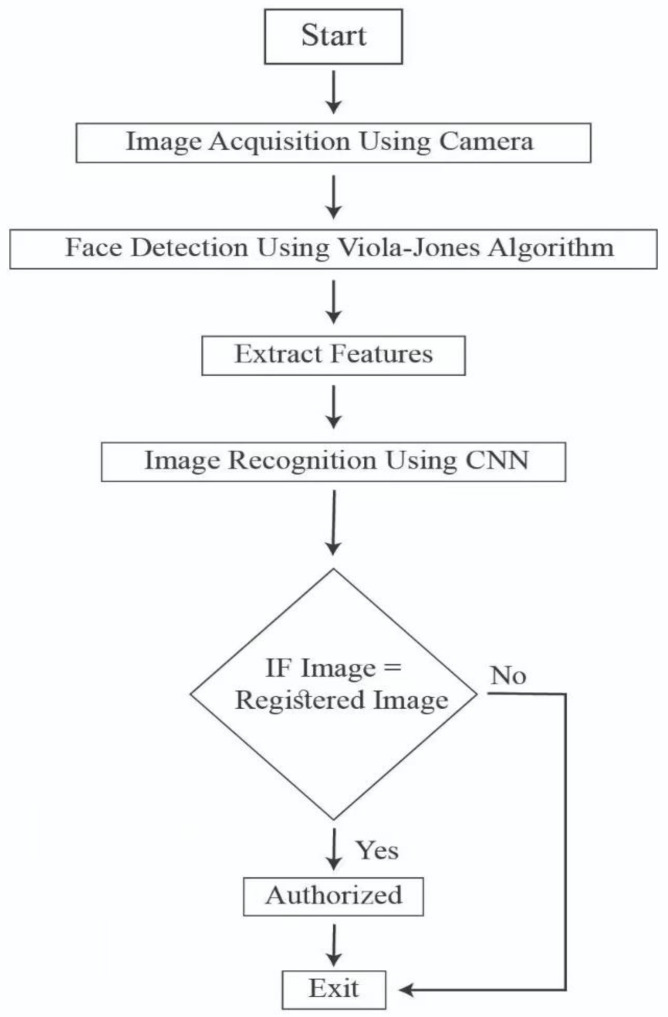
Image recognition module of screen monitoring.

**Figure 11 sensors-23-00275-f011:**
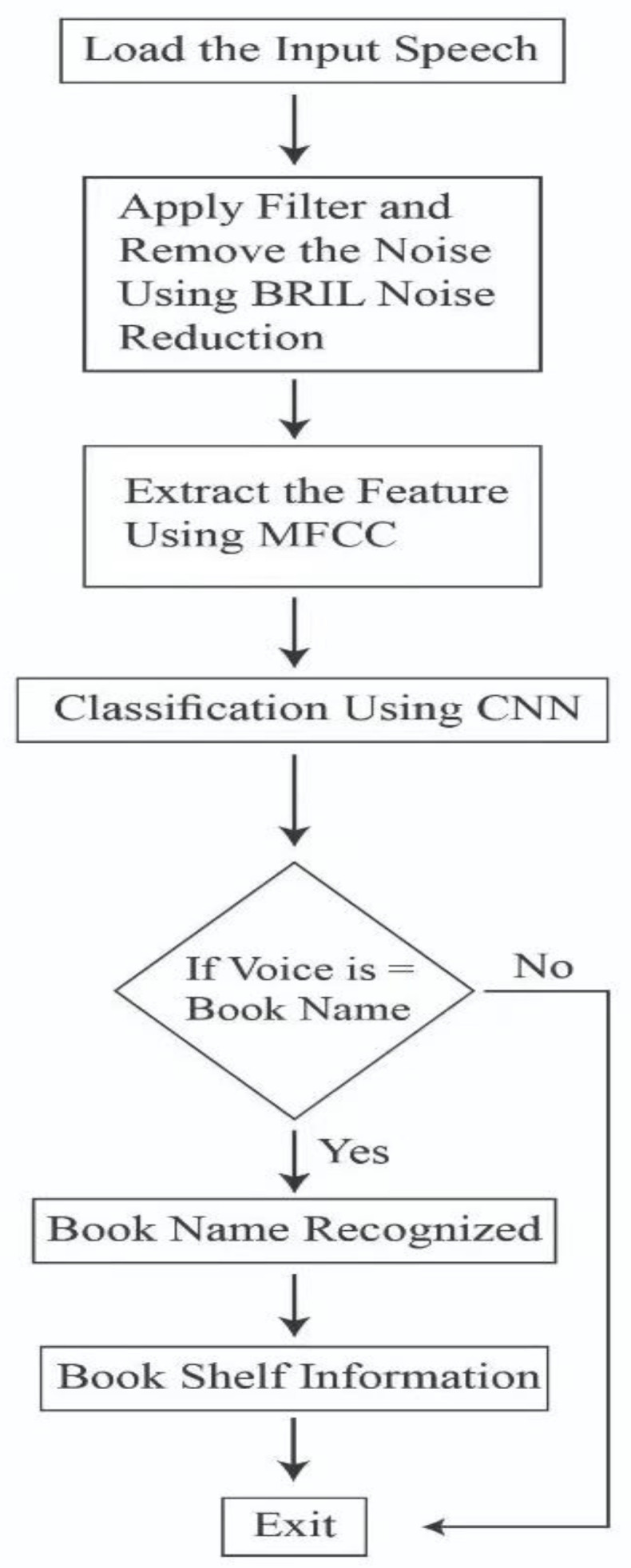
Voice recognition module of screen monitoring.

**Figure 12 sensors-23-00275-f012:**
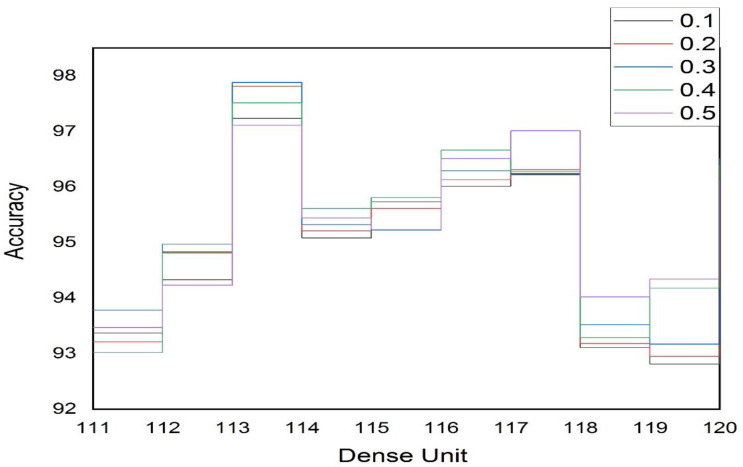
CNN Accuracy Dependent on the Density Unit and Dropout Rate for Facial Recognition.

**Figure 13 sensors-23-00275-f013:**
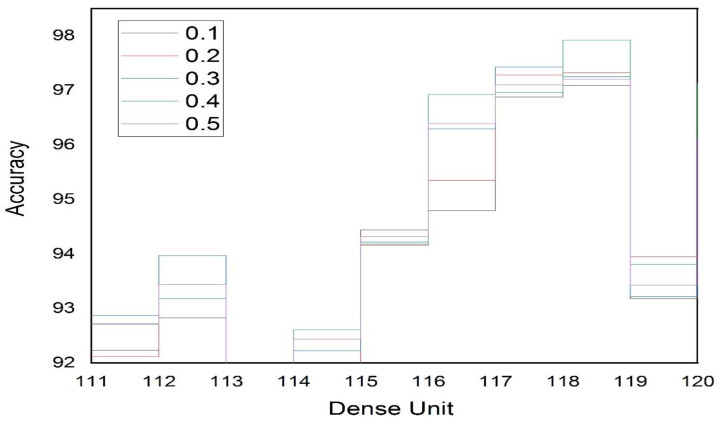
CNN Accuracy Dependent on the Density Unit and Dropout Rate for Voice Recognition.

**Figure 14 sensors-23-00275-f014:**
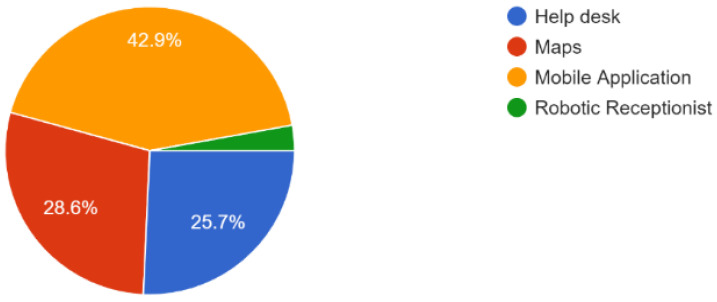
Available assistance for indoor navigation (35 Responses).

**Figure 15 sensors-23-00275-f015:**
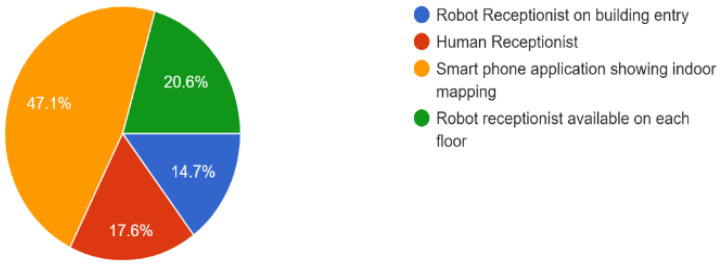
Responses of people for kind of help needed for indoor navigation in multistory building (35 Responses).

**Figure 16 sensors-23-00275-f016:**
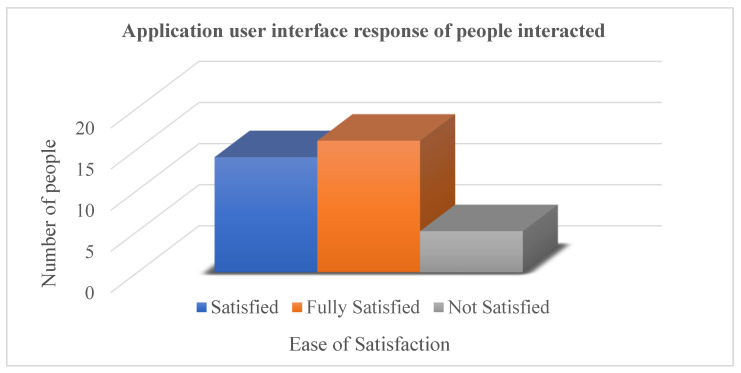
Application user interface response of people interacted (35 Responses).

**Figure 17 sensors-23-00275-f017:**
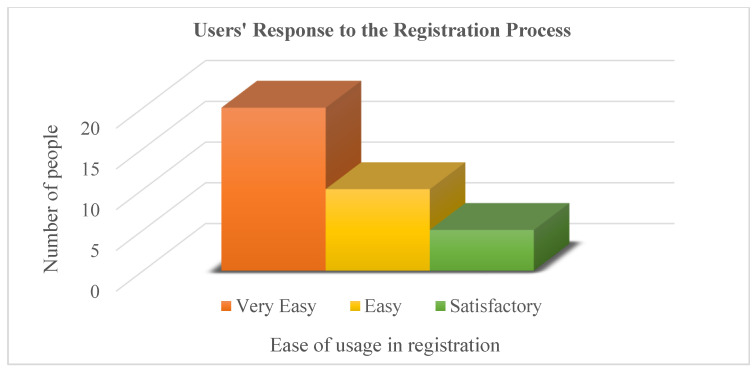
User registration difficulty level.

**Figure 18 sensors-23-00275-f018:**
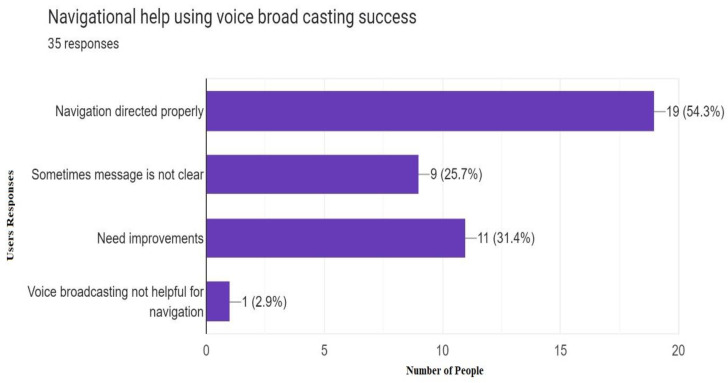
Response of accuracy of navigational help using voice broad casting (35 Responses).

**Figure 19 sensors-23-00275-f019:**
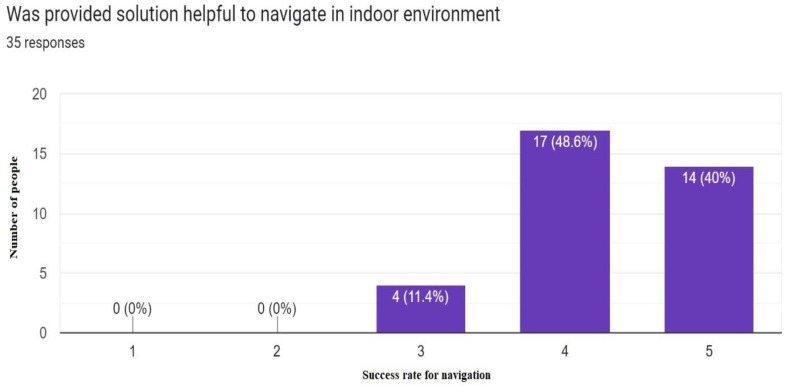
Success rate for navigation in indoor environment 5 being maximum (35 Responses).

**Figure 20 sensors-23-00275-f020:**
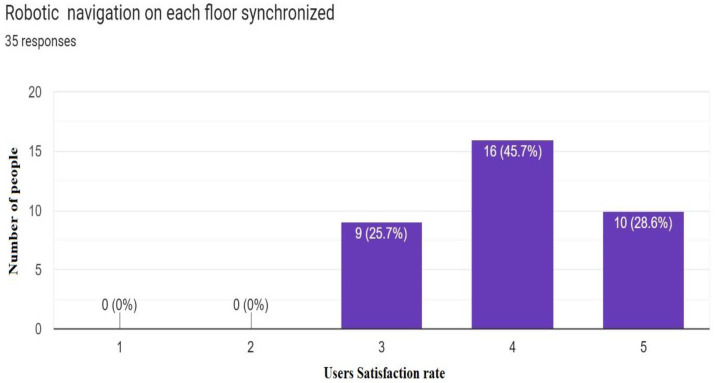
Synchronization of robots on each floor (35 Responses).

**Figure 21 sensors-23-00275-f021:**
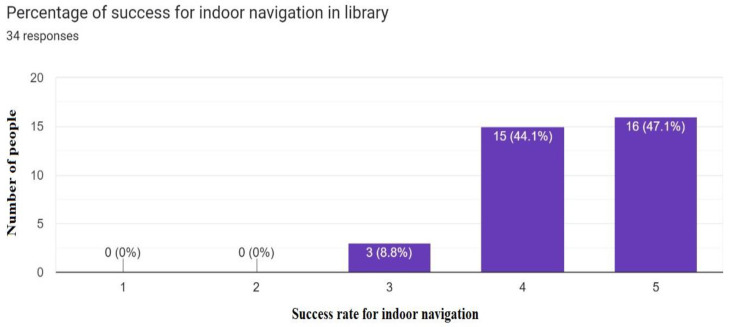
Success rate for indoor navigation in library.

**Figure 22 sensors-23-00275-f022:**
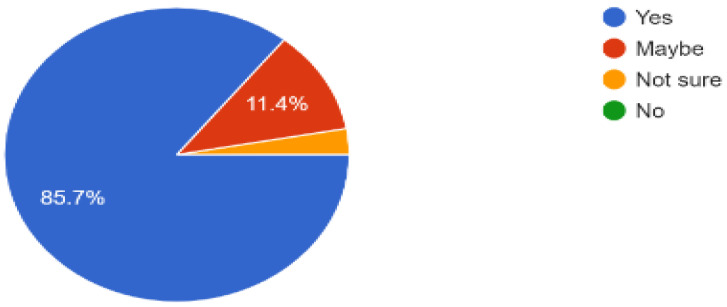
Future recommendation for installing in other multistory buildings.

**Figure 23 sensors-23-00275-f023:**
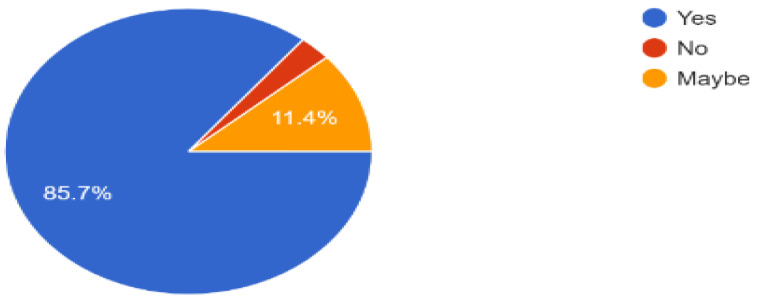
Responses to the question, can the provided solution can be implemented in other indoor multistory environments.

**Table 1 sensors-23-00275-t001:** Achieved Accuracy for Facial Recognition by using Dropout Rate and Density Unit.

Dropout Rate						
**Dense Unit**		**0.1**	**0.2**	**0.3**	**0.4**	**0.5**
	**111**	**93.37**	**93.21**	**93.78**	**93.02**	**93.47**
	**112**	**94.33**	**94.83**	**94.97**	**94.81**	**94.23**
	**113**	**97.23**	**97.81**	**97.88**	**97.51**	**97.11**
	**114**	**95.08**	**95.21**	**95.32**	**95.61**	**95.44**
	**115**	**95.73**	**95.61**	**95.22**	**95.81**	**95.23**
	**116**	**96.01**	**96.13**	**96.29**	**96.66**	**96.51**
	**117**	**96.22**	**96.31**	**96.24**	**96.27**	**97.01**
	**118**	**93.11**	**93.18**	**93.52**	**93.29**	**94.02**
	**119**	**92.81**	**92.95**	**93.17**	**94.18**	**94.34**
	**120**	**96.11**	**96.29**	**96.51**	**96.39**	**96.11**

**Table 2 sensors-23-00275-t002:** Achieved Accuracy for Voice Recognition by using Dropout Rate and Density Unit.

Dropout Rate						
**Dense Unit**		**0.1**	**0.2**	**0.3**	**0.4**	**0.5**
	**111**	**92.24**	**92.12**	**92.87**	**92.71**	**92.73**
	**112**	**92.83**	**93.44**	**93.97**	**93.18**	**93.44**
	**113**	**91.32**	**91.21**	**91.62**	**91.89**	**91.01**
	**114**	**90.08**	**90.91**	**92.23**	**92.61**	**92.44**
	**115**	**94.44**	**94.16**	**94.22**	**94.18**	**94.32**
	**116**	**94.79**	**95.35**	**96.29**	**96.92**	**96.39**
	**117**	**96.88**	**97.28**	**97.43**	**96.96**	**97.10**
	**118**	**97.09**	**97.32**	**97.25**	**97.92**	**97.20**
	**119**	**93.18**	**93.95**	**93.22**	**93.81**	**93.43**
	**120**	**96.21**	**96.89**	**97.01**	**97.13**	**96.11**

## Data Availability

All data is available in the manuscript.

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
