# Peer review of "An Intelligent Multi-Floor Navigational System Based on Speech, Facial Recognition and Voice Broadcasting Using Internet of Things"

_sensors, 2022, doi:10.3390/s23010275_

Round 1

Reviewer 1 Report

The paper proposes a solution based on navigational indoor system to be used to provide a specific location. The work consists of the development a mobile application to find a specific book in a library using an indoor navigation system, which can be adapted in many other final applications.

The work is very nice, the paper is very well written, which facilitates the reading.

The system proposed a solution to guide the user on each floor of a library. The solution is a mobile application that uses modern methods to recognize and authenticate the user, such as face and speech recognition, and also voice broadcasting to assist the user.

The main contribution of this solution is that several scopes are mixed: indoor navigational systems, indoor mapping, face and voice detection and recognition.

The main drawback of the solution is that the information of location has to be load to the system. In future applications, a generic indoor navigation system should be used.

All the work is very well-exposed, the methodology is described properly, and the conclusions have been coherently reasoned.

The references are appropriate, and the Literature State Section is quite well. It is not excessively extensive; it is quite concrete that facilitates the reading.

The figures are correct, but the name of the axis and the units must be indicated in all the graphs.

-        Figure1: X axis, name and units are missing

-        Figures 16, 17, 18, 19, 20, 21, 22, 23: X and Y axis, name and units are missing

I recommend the work for publication.

The only comment is about the structure of the paper, I think that the Literature Review section should belong to the Introduction Section, and the last part of the Introduction Section (from Line 77) should belong to the Methodology Section. It is only a suggestion, the paper in the present form is quite well.

Author Response

Please find the review comments in the attachment.

Reviewer 2 Report

1. This paper introduces the navigation system, but the space of the navigation function is too small, so it is suggested to add a detailed introduction to the navigation function.

2. The second half of the paper is less substantial than the beginning, so it is suggested to add more technical details.

3. The topic of the paper is relatively novel, but the detailed introduction of the function is relatively ordinary, so it is suggested to further improve the innovation of the paper.

4. A detailed comparison between the proposed method and other related methods in various aspects is suggested to further highlight the superiority of the proposed method.

5. The aspect ratio of some images in the paper is obviously abnormal, and the scaling ratio of the picture should be adjusted.

6. In terms of references, it is suggested to increase papers in relevant fields that are representative in recent years.

Author Response

Please find the review in the attachment.

Round 2

Reviewer 2 Report

The paper has been revised according to the review comments, and the questions have been answered clearly. It is recommended to be accepted.